# BiSPN: Generating Entity Set and Relation Set Coherently in One Pass

**Yuxin He**[1] and **Buzhou Tang**[1,2,*]

[1]Department of Computer Science, Harbin Institute of Technology, Shenzhen, China
[2]Peng Cheng Laboratory, Shenzhen, China
21S051047@stu.hit.edu.cn
tangbuzhou@gmail.com

## Abstract

By modeling the interaction among instances and avoiding error propagation, Set Prediction Networks (SPNs) achieve state-of-the-art performance on the tasks of named entity recognition and relation triple extraction respectively. However, how to jointly extract entities and relation triples via SPNs remains an unexplored problem, where the main challenge is the maintenance of coherence between the predicted entity/relation sets during one-pass generation. In this work, we present Bipartite Set Prediction Network (BiSPN), a novel joint entity-relation extraction model that can efficiently generate entity set and relation set in parallel. To overcome the challenge of coherence, BiSPN is equipped with a novel bipartite consistency loss as well as an entity-relation linking loss during training. Experiments on three biomedical/clinical datasets and a general-domain dataset show that BiSPN achieves new state of the art in knowledge-intensive scene and performs competitively in general-domain, while being more efficient than two-stage joint extraction methods.

## 1 Introduction

Extracting entities and relation triples from text is a fundamental task of Information Extraction. There have been many efforts that decompose the problem into separate tasks, i.e. named entity recognition (NER) and relation triple extraction (RE), and solve them respectively. Among these efforts, Set Prediction Networks (SPNs) have demonstrated state-of-the-art performance on NER (Tan et al., 2021; Shen et al., 2022) and RE (Sui et al., 2020; Tan et al., 2022).

Typically, SPNs leverage a set of learnable queries to model the interaction among instances (entities or relation triples) via attention mechanism and generate the set of instances naturally. The success of SPNs on NER and RE inspires us to explore

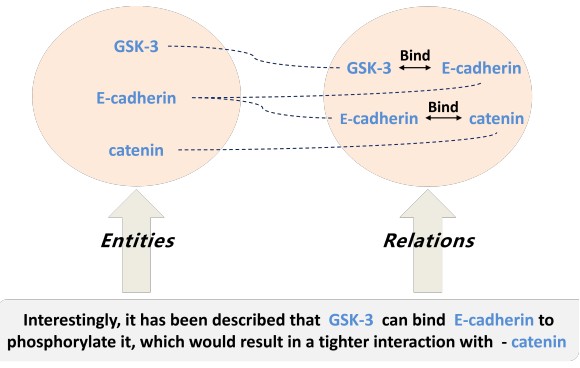

Figure 1: The target output of joint entity and relation extraction is essentially an entity set and a relation set that should be consistent with each other. Such coherence is difficult to be guaranteed when generating entity/relation sets in parallel. And our work manages to address this challenge.

the possibility of jointly solving the extraction of entities and relation triples with SPNs, which is a promising but unexplored direction.

In this paper, we propose Bipartite Set Prediction Network (BiSPN), a variant of SPNs, to generate target entity set and relation set in one pass. It can not only avoid the negative effect of cascade error but also enjoy the benefit of high inference speed. However, it is challenged by the difficulty to maintain the coherence between the generated entity set and relation set, due to its parallel design.

As illustrated in Figure 1, the head/tail entities of generated relation triples should be included in the generated entity set. The significance of this coherence is two-fold: 1) by requiring the generated entity set to contain the head/tail entities, the recall of the generated entities is more guaranteed; 2) by restricting the head/tail entities within the generated entity set, the precision and recall of the generated triples are more guaranteed.

Despite that, it is difficult to maintain such consistency when generating the two sets in parallel, since all instance queries are assumed to be of equal status and their hidden representations are updated

---

*Corresponding Author.

using bidirectional attention without any further restriction.

To overcome this challenge, we come up with two novel solutions. The first one is a *Bipartite Consistency Loss* function. It works by looking for a reference entity from the generated entity set for each relational subject/object and forcing the subject/object to simulate the reference entity. Symmetrically, it also find a reference subject/object for each entity classified as involved in relation and force the entity to simulate the reference subject/object. Our second solution is an *Entity-Relation Linking* Loss function, which works in the hidden semantic space. By computing the linking scores between the projected representations of entity queries and relation queries, it encourages the model to learn the interaction between entity instances and relation triple instances.

To sum up, our main contributions include:

- We present BiSPN, the first SPN-based joint entity and relation extraction model, which is able to generate target entity set and relation set in one pass.

- To maintain the coherence between the generated entity set and relation set, two novel loss functions, i.e., Bipartite Consistency Loss and Entity-Relation Linking Loss are introduced.

- BiSPN outperforms SOTA methods on three biomedical/clinical datasets, which is knowledge-intensive, and achieves competitive results on a general-domain benchmark. Besides, it infers much faster than two-stage joint extraction methods.

## 2 Related Work

### 2.1 Joint Entity and Relation Extraction

The target of joint entity and relation extraction is to recognize all entities mentioned in a given text and identify all entity pairs involved in relation. Existing methods for joint entity and relation extraction fall into four categories: (1) span-based methods (Dixit and Al-Onaizan, 2019; Zhong and Chen, 2021; Ye et al., 2022) that enumerate spans and conduct span-level classification to extract entities, enumerate and classify span pairs to extract relation triples; (2) table filling-based methods (Wang et al., 2020, 2021; Yan et al., 2021) that fill out a table for each entity/relation type via token pair classification; (3) machine reading comprehension

(MRC)-based methods (Li et al., 2019) that casts the task as a multi-turn question answering problem via manually designed templates; (4) autoregressive generation-based methods (Zeng et al., 2018; Lu et al., 2022) that reformulate the task as a sequence-to-sequence problem by linearizing target entities/relations into a pointer sequence or augmented natural language.

Among them, only the methods based on table filling can extract entities and relation triples in one stage, all other methods perform multi-step prediction, suffering from cascade errors and low inference speed. In this paper, we provide a new choice for one-stage joint entity and relation extraction, which is based on set prediction networks.

### 2.2 Set Prediction Networks

Set prediction networks are originally proposed for the task of object detection in computer vision (Carion et al., 2020). And they are successfully extended for information extraction by (Sui et al., 2020; Tan et al., 2021; Shen et al., 2022; Tan et al., 2022).

Generally, these methods employ a set of learnable queries as additional input, model the interaction among instances (entities/relations) via self-attention among the queries and one-way attention between the queries and the textual context. However, all these methods can only perform named entity recognition (Tan et al., 2021; Shen et al., 2022) or relation triple extraction (Sui et al., 2020; Tan et al., 2022), rather than jointly solve both of them.

## 3 Methodology

### 3.1 Problem Formulation

Given an input sentence $\mathbf{x} = x_1, x_2, ..., x_L$, the aim of joint entity and relation extraction is to predict the set of entities $\{e_i\}_{i=1}^{N_e}$ and the set of relation triples $\{r_j\}_{j=1}^{N_r}$ mentioned in the sentence. Here, $L$ is the length of the input sentence, $N_e$ and $N_r$ are the numbers of target entities and target relation triples respectively. The $i$-th entity $e_i$ is denoted as $(start_i, end_i, t_i^e)$, where $start_i, end_i$ are the start token index and end token index of the entity, $t_i^e \in \mathcal{T}_e$ is the entity type label. The $j$-th relation triple $r_j$ is denoted as $(e_j^h, t_j^r, e_j^t)$, where $e_j^h$ and $e_j^t$ are the head entity $(start_j^h, end_j^h, t_j^{e,h})$ and tail entity $(start_j^t, end_j^t, t_j^{e,t})$ of the relation triple, $t_j^r \in \mathcal{T}_r$ is the relation type label. We additionally define a null label $\varnothing$ for the set of entity types $\mathcal{T}_e$ and the

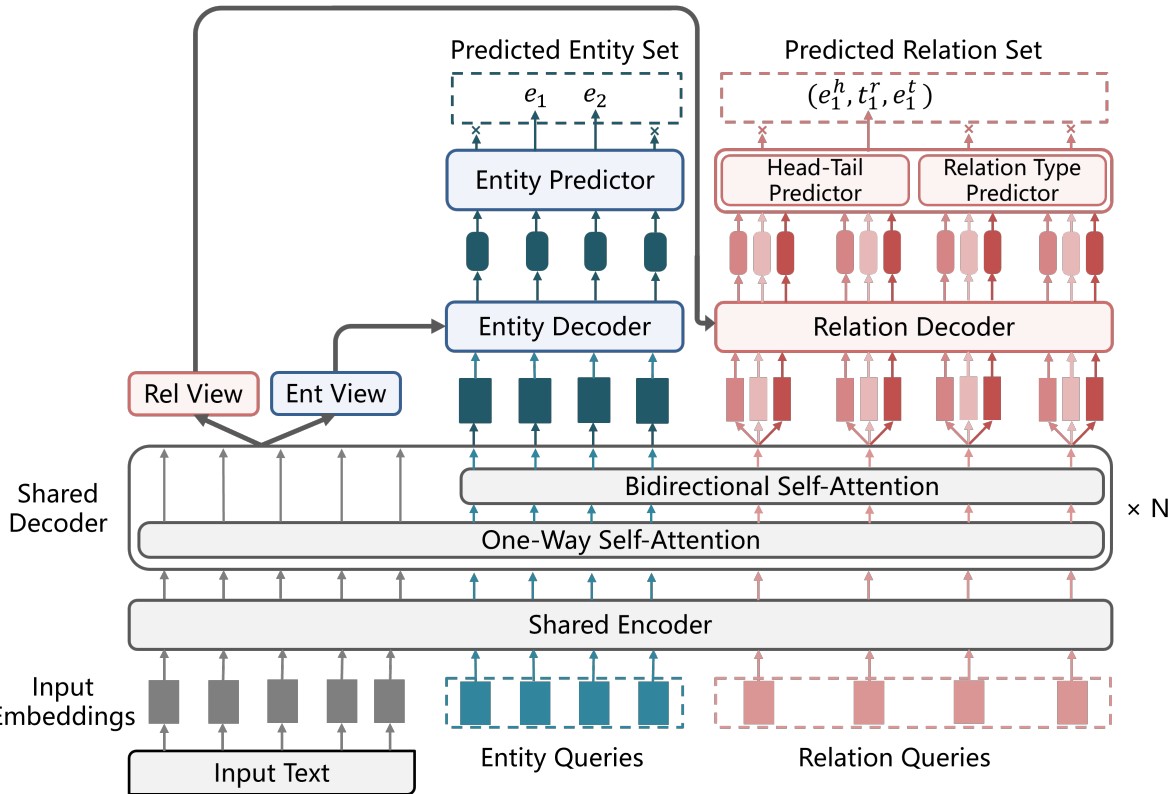

Figure 2: An overview of BiSPN, the proposed joint entity and relation extraction framework that is capable of generating target entity set and relation set coherently in one pass.

set of relation types $\mathcal{T}_r$ respectively, to indicate that no entity or no relation is recognized.

## 3.2 BiSPN

As illustrated in Figure 2, the proposed model mainly consists of a shared encoder, a shared decoder, an entity decoding module (in blue) and a relation decoding module (in red).

### 3.2.1 Shared Encoder

The encoder of BiSPN is essentially a bidirectional pretrained language model (Devlin et al., 2019) with modified input style and attention design.

We first transform the input sentence $\mathbf{x}$ into input embeddings $X \in \mathbb{R}^{L \times d}$, and then concatenate $X$ with a series of learnable entity queries $Q_e$ and relation queries $Q_r$ to form the model input $\tilde{X}$:

$$\tilde{X} = [X; Q_e; Q_r] \in \mathbb{R}^{(L+M_e+M_r) \times d} \quad (1)$$

where $d$ is the model dimension, $M_e$ and $M_r$ are hyperparameters controlling the number of entity queries and the number of relation queries ($M_e \gg N_e$, $M_r \gg N_r$).

To prevent the randomly initialized queries from negatively affecting the contextual token encodings,

we follow the work (Shen et al., 2022) to modify the bidirectional self-attention into **one-way self-attention**. Concretely, the upper right $L \times (M_e + M_r)$ sub-matrix of the attention mask is filled with negative infinity value so that the entity/relation queries become invisible to the token encodings, while the entity/relation queries can still attend to each other and the token encodings.

After multiple one-way self-attention layers and feed-forward layers, the encoder outputs the contextual token encodings as well as the contextual entity/relation queries.

### 3.2.2 Shared Decoder

The shared decoder consists of $N$ decoding blocks. Each decoding block includes an one-way self-attention layer (as described above), a bidirectional self-attention layer and feed-forward layers. The one-way self-attention layer here functions as the cross-attention layer of Transformer decoder (Vaswani et al., 2017), which aggregates textual context for decoding. (The main difference between one-way self-attention and cross-attention is that the contextual token encodings also get updated by one-way self-attention.) The bidirectional self-attention layer updates the entity/relation

queries via modeling the interaction among them.

After shared decoding, the decoder outputs the updated token representations $H^x$, entity queries $H^e$ and relation queries $H^r$.

### 3.2.3 Entity Decoding Module

The entity decoding module consists of an entity-view projection layer, an entity decoder and an entity predictor.

The entity-view projection layer first linearly transforms the token encodings $H^x$ into entity-view:

$$H_e^x = \text{Linear}(H^x) \tag{2}$$

The entity decoder, which includes multiple layers of cross-attention and bidirectional self-attention, receives the transformed token encodings $H_e^x$ as decoding context and the entity queries $H_e$ as decoder input, and output the final representation of entity queries $\tilde{H}^e$:

$$\tilde{H}^e = \text{EntityDecoder}(H^e|H_e^x) \tag{3}$$

The entity predictor is responsible for predicting the boundary and entity type of each entity query.

For each entity query, it first fuse the query representation with the transformed token encodings, and then calculate the probability of each token in the sentence being the start/end token of the corresponding entity:

$$S_i^\delta = \text{Linear}\big(\text{Relu}(\text{Linear}(\tilde{H}_i^e) + \text{Linear}(H_e^x))\big) \tag{4}$$

$$P_i^\delta = \text{Softmax}(S_i^\delta),\ \delta \in \{\text{start}, \text{end}\} \tag{5}$$

where $S_i^\delta \in \mathbb{R}^L$ is a vector of logits of each token being the start/end token of the entity associated with $i$-th entity query, $P_i^\delta$ is the corresponding probability distribution.

An MLP-based classifier is leveraged to predict the type of the entity associated with the $i$-th entity query:

$$P_i^{t^e} = \text{Softmax}(\text{MLP}(\tilde{H}_i^e)) \tag{6}$$

During inference, the predicted boundary and entity type corresponding to the $k$-th entity query are calculated as:

$$\text{score}_k(i, j) = P_k^{\text{start}}[i] + P_k^{\text{end}}[j] \tag{7}$$

$$(\hat{start}_k, \hat{end}_k) = \underset{(i,j):\ 0<j-i<L}{\arg\max}\ \text{score}_k(i, j) \tag{8}$$

$$\hat{t}_k^e = \arg\max P_k^{t^e} \tag{9}$$

Note that, the entity predictor will filter out the entity whose predicted type label is $\varnothing$.

### 3.2.4 Relation Decoding Module

The relation decoding module consists of a relation-view projection layer, a relation decoder, a head-tail predictor and a relation type predictor.

The relation-view projection layer and relation decoder work in the same manner as the entity-view projection layer and entity decoder, except that the relation decoder splits relation queries into head/tail queries before decoding:

$$[H^h; H^t] = \text{Linear}(H^r) \tag{10}$$

$$H_r^x = \text{Linear}(H^x) \tag{11}$$

$$\tilde{H}^h, \tilde{H}^t, \tilde{H}^r = \text{RelationDecoder}(H^h, H^t, H^r|H_r^x) \tag{12}$$

The head-tail predictor then predicts the boundary and entity type of the head/tail entity associated with each relation queries. This process is similar to the entity prediction process (Equation 4-10). The only difference is that the entities queries becomes the head/tail queries $\tilde{H}^{h/t}$ and the token encodings is now in relation-view $H_r^x$.

The relation type predictor classifies the category of $i$-th relation query according to $\tilde{H}_i^r$:

$$P_i^{t^r} = \text{Softmax}(\text{MLP}(\tilde{H}_i^r)) \tag{13}$$

### 3.3 Prediction Loss

To train the model, we should find the optimal assignment between the gold entity set and the generated entity set, as well as the optimal assignment between the gold relation set and the generated relation set, which are calculated in the same way as in (Tan et al., 2021; Shen et al., 2022) using the Hungarian algorithm (Kuhn, 1955).

After the optimal assignments are obtained, we calculate the following prediction loss $\mathcal{L}_{\text{pred}}$ for each sample:

$$\mathcal{L}_{\text{ent}} = -\sum_{i=1}^{M_e} \big( \log P_i^{\text{start}}[start_{\varphi(i)}] + \log P_i^{\text{end}}[end_{\varphi(i)}]$$
$$+ \log P_i^{t^e}[t_{\varphi(i)}^e] \big) \tag{14}$$

$$\mathcal{L}_{\text{ent}}^{\text{h/t}} = -\sum_{j=1}^{M_r} \big( \log P_{j,\text{h/t}}^{\text{start}}[start_{\sigma(j)}^{\text{h/t}}] + \log P_{j,\text{h/t}}^{\text{end}}[end_{\sigma(j)}^{\text{h/t}}]$$
$$+ \log P_{j,\text{h/t}}^{t^e}[t_{\sigma(j)}^{e,\text{h/t}}] \big) \tag{15}$$

$$\mathcal{L}_{\text{rel}} = \mathcal{L}_{\text{ent}}^{\text{h/t}} - \sum_{j=1}^{M_r} \log P_j^{t^r}[t_{\sigma(j)}^r] \tag{16}$$

$$\mathcal{L}_{\text{pred}} = \mathcal{L}_{\text{ent}} + \mathcal{L}_{\text{rel}} \tag{17}$$

where $\varphi(i)$ is the index of the gold entity assigned to the $i$-th generated entity, $\sigma(j)$ is the index of the

gold relation triple assigned to the $j$-th generated relation triple, $\mathcal{L}_{\text{ent}}^{\text{h/t}}$ represents the loss of head/tail entity prediction.

## 3.4 Bipartite Consistency Loss

To calculate the bipartite consistency loss, we first find a reference entity from the generated entity set for each head/tail entity. A reference entity is defined as the entity most similar to the referring head/tail entity. Concretely, the similarity between $e_a$, the $a$-th generated entity, and $e_b^{\text{h/t}}$, the head/tail entity of the $b$-th generated relation triple is measured in KL divergence between the start/end/type probability distributions of $e_a$ and $e_b^{\text{h/t}}$:

$$\text{sim}(e_a, e_b^{\text{h/t}}) = \sum_{\delta \in \{\text{start,end},t^e\}} -D_{\text{KL}}(P_a^\delta || P_b^{\delta,\text{h/t}})$$

where $D_{\text{KL}}(P||Q)$ is the KL divergence between target distribution $P$ and approximate distribution $Q$; $P_b^{\delta,\text{h/t}}$ means $P_b^{\delta,h}$ when $e_b^{\text{h/t}}$ is a head entity, otherwise $P_b^{\delta,\text{h/t}}$ means $P_b^{\delta,t}$.

We want every head/tail entity to simulate its reference entity, which is equivalent to maximizing the similarity. Hence, the consistency loss in the **relation → entity** direction is computed as:

$$\mathcal{L}_{\text{rel}\to\text{ent}} = -\sum_{i=1}^{M_r} \Big[ \max_{j\in(1,M_e)} \text{sim}(e_j, e_i^h) \\ + \max_{j\in(1,M_e)} \text{sim}(e_j, e_i^t) \Big] \quad (18)$$

Symmetrically, we also find a reference head/tail entity for each generated entity that is classified as having relation. The classification is conducted by a binary classifier, which is trained with a binary cross-entropy loss function:

$$\mathbf{p}_i^{\text{has-rel}} = \text{sigmoid}(\text{MLP}(\tilde{H}_i^e)) \quad (19)$$

$$\mathcal{L}_{\text{has-rel}} = -\frac{1}{M_e}\sum_{i=1}^{M_e} \big[ y_i^{\text{has-rel}} \log \mathbf{p}_i^{\text{has-rel}} + \\ (1-y_i^{\text{has-rel}})\log(1-\mathbf{p}_i^{\text{has-rel}})\big] \quad (20)$$

where $y_i^{\text{has-rel}} = 1$ only if the gold entity assigned to the $i$-th entity query is involved in relation.

The consistency loss in the **entity → relation** direction is then calculated as follows:

$$\tilde{\text{sim}}(e_b^{\text{h/t}}, e_a) = \sum_{\delta \in \{\text{start,end},t^e\}} -D_{\text{KL}}(P_b^{\delta,\text{h/t}} || P_a^\delta)$$

$$\mathcal{L}_{\text{ent}\to\text{rel}} = -\sum_{i\in\Omega} \max_{j\in(1,M_r)} \tilde{\text{sim}}(e_j^{\text{h/t}}, e_i) \quad (21)$$

$$\Omega = \{i \mid \mathbf{p}_i^{\text{has-rel}} \geq 0.5, i \in (1, M_e)\} \quad (22)$$

where $\Omega$ is the set of indices of the entities classified as involved in relation.

We sum up $\mathcal{L}_{\text{ent}\to\text{rel}}$, $\mathcal{L}_{\text{rel}\to\text{ent}}$ and $\mathcal{L}_{\text{has-rel}}$ to get the overall bipartite consistency loss $\mathcal{L}_{\text{ent}\leftrightarrow\text{rel}}$.

## 3.5 Entity-Relation Linking Loss

While the bipartite consistency loss softly aligns the predicted distributions between the generated entity set and relation set during training, the entity-relation linking loss encourages BiSPN to model the interaction between entity queries and relation queries.

To this end, we first project the intermediate representations of entity queries and relation queries and then compute the linking scores between them via a Biaffine layer:

$$\bar{H}^e = \text{Linear}(H^e) \quad (23)$$
$$\bar{H}^r = \text{Linear}(H^r) \quad (24)$$
$$S^{\text{link}} = \text{Biaffine}(\bar{H}^e, \bar{H}^r) \in \mathbb{R}^{M_e \times M_r} \quad (25)$$

With the linking scores, we calculate the following binary cross-entropy loss:

$$P^{\text{link}} = \text{sigmoid}(S^{\text{link}}) \quad (26)$$

$$\mathcal{L}_{\text{link}} = -\frac{1}{M_e M_r}\sum_{i=1}^{M_e}\sum_{j=1}^{M_r} \big[ y_{i,j}^{\text{link}} \log P_{i,j}^{\text{link}} \\ + (1-y_{i,j}^{\text{link}})\log(1-P_{i,j}^{\text{link}})\big] \quad (27)$$

where $y_{i,j}^{\text{link}} = 1$ only if the gold entity assigned to the $i$-th entity query appears in the gold relation triple assigned to the $j$-th relation query.

# 4 Experiments

## 4.1 Experiment Setups

| Statistics | ACE05 | BioRelEx | ADE | Text2DT |
|---|---|---|---|---|
| # Sentences | 14525 | 2010 | 4272 | 500 |
| Avg sent. length | 14.6 | 29.0 | 20.3 | 66.5 |
| # Entity types | 7 | 33 | 2 | 6 |
| # Relation types | 6 | 3 | 1 | 6 |
| # Entities | 38287 | 9871 | 17808 | 3280 |
| # Relations | 7691 | 3235 | 6821 | 3196 |
| # Ent. per sent. | 2.63 | 4.91 | 4.17 | 6.56 |
| # Rel. per sent. | 0.53 | 1.61 | 1.60 | 6.39 |

Table 1: Statistics of the ACE05, BioRelEx, ADE and Text2DT datasets.

| Paradigm | Method | ACE05 | | BioRelEx | | ADE | | Text2DT | |
|---|---|---|---|---|---|---|---|---|---|
| | | Ent-F1 | Rel-F1 | Ent-F1 | Rel-F1 | Ent-F1 | Rel-F1 | Ent-F1 | Rel-F1 |
| Two-stage | KECI (2021) | - | - | 87.4 | 66.1 | 90.7 | 81.7 | - | - |
| | MADR (2023) | - | - | - | - | 91.8 | 80.1 | - | - |
| | PL-Marker (2022) | **89.8** | **66.5** | 87.4 | 66.3 | 92.2 | 83.2 | 96.3 | 93.3 |
| One-stage | PFN (2021) | 88.4 | 64.9 | 87.1 | 65.8 | 91.3 | 83.2 | 95.1 | 92.9 |
| | UniRE (2021) | 88.8 | 64.3 | 87.3 | 65.6 | 91.6 | 82.9 | 95.4 | 92.5 |
| | TOP1 (2021; 2022) | - | - | - | - | - | - | - | 94.4 |
| | BiSPN (Ours) | 89.7 | 65.5 | **87.5** | **67.0** | 92.1 | **83.7** | **97.1** | **94.9** |
| | w/o $\mathcal{L}_{ent\leftrightarrow rel}$ | 89.2 | 64.0 | 86.1 | 66.2 | 91.4 | 83.5 | 96.8 | 94.4 |
| | w/o $\mathcal{L}_{link}$ | 89.5 | 65.1 | 87.4 | 66.7 | 91.8 | 83.5 | **97.1** | 94.6 |
| | w/o $\mathcal{L}_{ent\leftrightarrow rel}$, $\mathcal{L}_{link}$ | 88.9 | 63.4 | 85.7 | 66.0 | 90.9 | 83.0 | 96.2 | 94.1 |

Table 2: Main results on the ACE05, BioRelEx, ADE and Text2DT datasets. The bold font indicates the best score and the underline font indicates the second-best score.

**Datasets.** We experiment on one general-domain dataset (ACE05) and three knowledge-intensive datasets (BioRelEx, ADE, Text2DT). **ACE05** (Walker et al., 2006) includes a corpus of newswire, broadcast news, telephone conversations. **BioRelEx** (Khachatrian et al., 2019) contains biomedical literature about binding interactions between proteins and/or biomolecules. **ADE** (Gurulingappa et al., 2012) consists of medical reports describing drug-related adverse effects. **Text2DT**[1] is originally an benchmark for medical **D**ecision **T**rees extraction task in China Health Information Processing Conference 2022. And we only use its entity/relation annotation for experiments. See Table 1 for detailed statistics of the datasets.

We additionally experiment on the **SciERC** dataset, where we follow the same setting as in (Wang et al., 2021; Ye et al., 2022). See Appendix B for the results.

**Evaluation Metrics.** Strict evaluation metrics are applied, where an entity is confirmed correct only if its boundary and entity type are correctly predicted; a relation triple is confirmed correct only if its relation type and head/tail entity are correctly predicted. For ACE05 and BioRelEx, we report the averaged Micro F1 scores over 3 random seeds. For ADE, we follows (Ji et al., 2020; Lai et al., 2021) to conduct 10-fold cross-validation and report the averaged Macro F1 scores. For Text2DT, we follows the top-1 system on the evaluation task to ensemble 5 models trained with different random seeds and report the Micro F1 scores.

[1] http://www.cips-chip.org.cn/2022/eval3

| Method | ACE05 | | BioRelEx | |
|---|---|---|---|---|
| | Rel-F1 | sent/s | Rel-F1 | sent/s |
| KEIC (2021) | - | - | 66.1 | 15.7 |
| UniRE (2021) | 64.3 | **134.6** | 65.8 | **107.2** |
| PL-Marker (2022) | **66.5** | 32.0 | 66.4 | 21.9 |
| BiSPN (Ours) | 65.5 | 59.7 | **67.0** | 40.8 |

Table 3: Inference speed comparison on the ACE05, BioRelEx datasets.

## 4.2 Implementation Details

We implement BiSPN with Pytorch (Paszke et al., 2019) and run experiments on NVIDIA Tesla V100 GPUs. For ACE05, we follow (Wang et al., 2021; Ye et al., 2022) to initialize the shared encoder with BERT-base (Devlin et al., 2019). For BioRelEx and ADE, we follow (Haq et al., 2023; Lai et al., 2021) to initialize the shared encoder with BioBERT-base. For Text2DT, we initialize the shared encoder with Chinese-bert-wwm-ext (Cui et al., 2021). The decoding modules are randomly initialized. Following (Shen et al., 2022), we freeze the encoder in the first 5 epochs and unfreeze it in the remaining epochs. The learning rate of decoding modules is set to be larger than the learning rate of the encoder. We adopt an AdamW optimizer (Loshchilov and Hutter, 2017) equipped with a linear warm-up scheduler to tune the model. See Appendix A for details of hyperparameter tuning.

## 4.3 Compared Baselines

We compare BiSPN with several SOTA methods listed as follows.

**KECI** (Lai et al., 2021): A knowledge-enhanced two-stage extraction model based on span graphs.

**MADR** (Haq et al., 2023): A pipeline of independent NER and RE models.

**PL-Marker** (Ye et al., 2022): A span-based method that models the interrelation between spans by packing levitated markers in the encoder.

**PFN** (Yan et al., 2021): A partition filter network that models two-way interaction between NER and RE subtasks.

**UniRE** (Wang et al., 2021): A method based on table filling, featured with a unified label space for one-stage joint entity and relation extraction.

**TOP1**: The top-1 system on the Text2DT evaluation task, which combines PFN (Yan et al., 2021) with Efficient GlobalPointer (Su, 2022).

Note that, we do not compare with SPN (Sui et al., 2020), UniRel (Tang et al., 2022) and QIDN (Tan et al., 2022), since these methods can only extract relation triples and cannot recognize those entities uninvolved in relation.

### 4.4 Main Results

Table 2 summarizes the overall performance of BiSPN and compared baselines on ACE05, BioRelEx, ADE and Text2DT. In terms of entity extraction, BiSPN performs competitively with or slightly better than SOTA methods on ACE05, BioRelEx and ADE, while outperforming the SOTA method PL-Marker by 0.8 F1 on Text2DT.

In terms of relation extraction, BiSPN boosts SOTA performance by 0.7, 0.5 and 0.5 F1 on BioRelEx, ADE and Text2DT respectively, verifying the effectiveness of our method on knowledge-intensive scene. However, although BiSPN outperforms SOTA one-stage methods by 0.6 F1 on ACE05, it is no match for the two-stage method PL-Marker on this general-domain dataset. We will look into the reason behind this in Section 4.6.2.

### 4.5 Inference Efficiency

We compare the inference efficiency of KEIC, UniRE, PL-Marker, BiSPN on the BioRelEx and Text2DT datasets. For a fair comparison, the experiments are all conducted on a server with Intel(R) Xeon(R) E5-2698 CPUs and NVIDIA Tesla V100 GPUs. And we fix the batch size as 8 during evaluation. As shown in Table 3, BiSPN can process around 40~60 sentences per second, which is 2 times faster than the SOTA two-stage method PL-Marker. Although UniRE, a SOTA one-stage method, is about 2.5 times faster than BiSPN, its performance of relation extraction is uncompetitive against BiSPN.

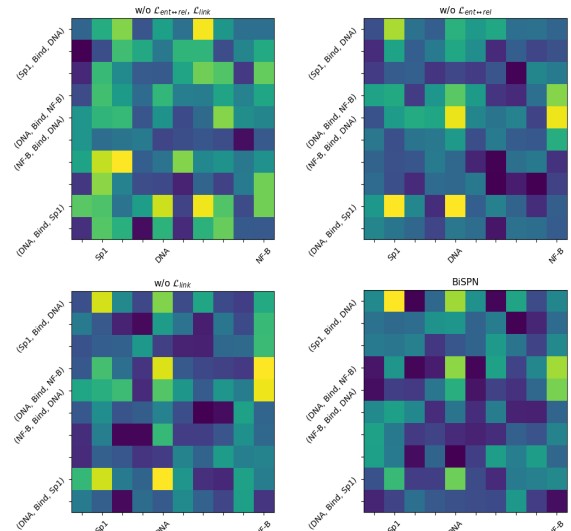

Figure 3: Visualization of attention between entity queries and relation queries of a sample from BioRelEx. The input text in this sample is "Moreover, the in vitro binding of NF-B or Sp1 to its target DNA was not affected by the presence of K-12".

| Method | # Rel $\geq$ 0 [10051, 2050] | # Rel $\geq$ 1 [2643, 597] | # Rel $\geq$ 2 [1203, 302] |
|---|---|---|---|
| PL-Marker | 66.5 | 63.8 | 57.9 |
| BiSPN | 65.5 (-1.0) | 63.4 (-0.4) | 58.2 (+0.3) |

Table 4: Relation F1 scores of BiSPN on ACE05 with different settings of knowledge density. The numbers in each bracket are the numbers of training, testing samples in the dataset under specific setting.

### 4.6 Analysis

#### 4.6.1 Effects of $\mathcal{L}_{ent \leftrightarrow rel}$ and $\mathcal{L}_{link}$

We conduct ablation study and qualitative visualization to analyze how the bipartite consistency loss $\mathcal{L}_{ent \leftrightarrow rel}$ and entity-relation linking loss $\mathcal{L}_{link}$ work.

The results of ablation study are shown in Table 2. Without the bipartite consistency loss, the entity F1 scores drop by 0.5, 1.4, 0.7, 0.3 and the relation F1 scores drop by 1.5, 0.8, 0.2, 0.5 on ACE05, BioRelEx, ADE and Text2DT respectively. Without the entity-relation linking loss, the entity F1 scores decrease by 0.2, 0.1, 0.3, 0 and the relation F1 scores decrease by 0.4, 0.3, 0.2, 0.3 on the four datasets respectively. After removing the bipartite consistency loss and entity-relation linking loss together, the entity F1 scores decrease by 0.8, 1.8, 1.2, 0.9 and the relation F1 scores decrease by 2.1, 1.0, 0.7, 0.8 on the four datasets respectively.

Three conclusions can be derived from these results: 1) both the bipartite consistency loss and

| Input Text | BiSPN | | w/o $\mathcal{L}_{ent\leftrightarrow rel}, \mathcal{L}_{link}$ | |
| --- | --- | --- | --- | --- |
| | Entity | Relation | Entity | Relation |
| North Korea has told American lawmakers it already has nuclear weapons and intends to build more, a senior U.S. | GPE: North Korea
GPE: American
PER: lawmakers
GPE: it    WEA: weapons
WEA: more  GPE: U.S. | (lawmakers, ORG-AFF, U.S) ×
(it, ART, weapons)
(it, ART, more)
(lawmakers, ORG-AFF, American) | GPE: North Korea
GPE: American
PER: lawmakers
GPE: it    WEA: weapons
GPE: U.S.
WEA: more | (lawmakers, ORG-AFF, U.S) ×
(it, ART, weapons)
(lawmakers, ORG-AFF, American)

(it, ART, more) |
| The general transcription factors TFIIA and TFIIB bind directly to TBP, stabilize its association with the promoter, and form the core of the RNA polymerase II preinitiation complex ( 18, 36, 48). | Protein: TFIIA
Protein: TFIIB
Protein: TBP
DNA: promoter
Protein-complex: RNA polymerase II
Protein-complex: RNA polymerase II preinitiation complex | (TBP, bind, TFIIA)
(TBP, bind, TFIIB)
(TBP, bind, promoter)
(TFIIA, bind, TBP)
(TFIIB, bind, TBP)
(promoter, bind, TBP) | Protein: TFIIA
Protein: TFIIB
Protein: TBP
Protein-complex: RNA polymerase II

DNA: promoter
Protein-complex: RNA polymerase II preinitiation complex | (TBP, bind, TFIIA)
(TBP, bind, TFIIB)
(TFIIA, bind, TBP)
(TFIIB, bind, TBP)

(TBP, bind, promoter)
(promoter, bind, TBP) |

Figure 4: Cases from ACE05 and BioRelEx. False negative predictions are in blue.

entity-relation linking loss contribute to the overall performance of BiSPN; 2) the bipartite consistency loss is much more effective than the entity-relation linking loss; 3) the effects of the two losses are not orthogonal but still complementary in some degree.

To visualize the attention between entity queries and relation queries, we record the attention weight in the last bidirectional self-attention layer of shared decoder for a sample from BioRelEx. Since the original weight is bidirectional, we average the weight over two directions and conduct normalization to obtain the weight for visualization. As shown in Figure 3, without $\mathcal{L}_{ent\leftrightarrow rel}$ and $\mathcal{L}_{link}$, the interaction between entity queries and relation queries is chaotic and pointless. After applying $\mathcal{L}_{ent\leftrightarrow rel}$ or $\mathcal{L}_{link}$ separately, the relation queries associated with gold relation triples tend to focus on the entity queries associated with gold entities. When applying the two loss functions together, the relation queries associated with gold relation triples further concentrate on the entity queries whose target entities are the same as their head/tail entities. This phenomenon coincides with the purpose of the two loss functions.

#### 4.6.2 Influence of Knowledge Density

The performance of BiSPN on ACE05 is not so good as its performance on BioRelEx, ADE and Text2DT. We hypothesize it is the sparsity of relation triples that hinders the learning of BiSPN. As listed in Table 1, the average number of relations per sentence is 0.53 on ACE05. In contrast, the average numbers of relations per sentence are 1.61, 1.60 and 6.39 on the other three datasets.

To verify our hypothesis, we filter samples according to the number of relations they contain and experiment on different versions of the filtered dataset. As shown in Table 4, when the samples

without relation are discarded, the performance gap between PL-Marker and BiSPN narrows from 1.0 to 0.4. When further discarding the samples with less than 2 relation triples, BiSPN even performs slightly better than PL-Marker. This reveals that the strength of BiSPN emerges in knowledge-intensive scene, which is reasonable, since BiSPN works by modeling interaction among knowledge instances.

### 4.7 Case Study

Figure 4 illustrates two test cases from ACE05 and BioRelEx respectively. In the first case, BiSPN without $\mathcal{L}_{ent\leftrightarrow rel}$ and $\mathcal{L}_{link}$ fails to recognizes the WEA entity "more" and the PART-WHOLE relation between "it" and "more", while BiSPN successfully extracts them by considering the context in entity-view and relation-view concurrently. Likewise, in the second case, BiSPN successfully recognizes the entity "promoter" after $\mathcal{L}_{ent\leftrightarrow rel}$ and $\mathcal{L}_{link}$ is applied. However, it still fails to recognize "RNA polymerase II preinitiation complex", which is complicated and may require domain knowledge for recognition.

## 5 Conclusion

In this work, we present BiSPN, a novel joint entity relation extraction framework based on bipartite set prediction. It generates entity set and relation set in a distributed manner, so as to avoid error propagation. To maintain the coherency between the generated entity set and relation set, we come up with two novel loss designs, namely bipartite consistency loss and entity-relation linking loss. The first one pulls closer the predicted boundary/type distributions of entities and head/tail entities, while the second one enforces the interaction between entity queries and relation queries. Extensive experiments demonstrate the advantage of BiSPN in

knowledge-intensive scene, as well as the effectiveness of the proposed bipartite consistency loss and entity-relation linking loss.

## Limitations

As mentioned in Section 4.6.2, the performance of our BiSPN framework can be hindered when the distribution of relation triples are overly sparse in a dataset. This suggests that BiSPN is a better choice for biomedical and clinical domains but not the general-domain, where knowledge sparsity is common.

Another limitation of BiSPN is its reliance on a fixed number of entity/relation queries. Although it is possible to set the number of queries larger in order to let BiSPN generalize to longer text input, the cost is additional memory and time consumption that grows quadratically. To effectively address this, future work can draw lessons from the field of dynamic neural network and explore dynamic selection of instance queries.

## Acknowledgments

We thank the reviewers for their valuable suggestions. This study is partially supported by National Key R&D Program of China (2021ZD0113402 ), National Natural Science Foundation of China (62276082), Major Key Project of PCL (PCL2021A06), Shenzhen Soft Science Research Program Project (No.KX202207705152815035) and the Fundamental Research Fund for the Central Universities (HIT.DZJJ.2023117).

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

# A  Hyperparameter Configuration

We tune the hyperparameters for each dataset by manually trying different values of each hyperparameter within a specific interval and choosing the

value that results in the highest relation F1 on the development set (For ADE, the validation set of its first fold of data is employed as the development set). After trial, we find it optimal to set the numbers of shared decoder layers, entity decoder layers and relation decoder layers as 4, 1, 1 for all datasets. The trial intervals and final configuration of other hyperparameters are shown in Table 5.

## B Experiment Results on the SciERC dataset

Here, we append the results of additional experiment on the SciERC dataset. As shown in Table 6, in terms of entity recognition, BiSPN underperforms SOTA two-stage method PL-Marker by Ent-F1, but still outperforms SOTA one-stage methods (PFN, UniRE) substantially. In terms of relation triple extraction, BiSPN establishes new SOTA (Rel-F1) on the SciERC dataset. In terms of inference speed, BiSPN is about faster than PL-Marker, but slower than other one-stage methods. The results after ablating the consistency loss and the linking loss verify the effectiveness of them on the dataset.

| Parameter | Trial Interval | ACE05 | BioRelEx | ADE | Text2DT | SciERC |
|-----------|----------------|-------|----------|-----|---------|--------|
| Epochs | [50, 100] | 70 | 75 | 70 | 80 | 70 |
| Warmup Rate | [0.01, 0.2] | 0.1 | 0.1 | 0.1 | 0.1 | 0.1 |
| Encoder $lr$ | [1e-5, 5e-5] | 2e-5 | 2e-5 | 2e-5 | 2e-5 | 2e-5 |
| Decoder $lr$ | [1e-5, 5e-5] | 4e-5 | 4e-5 | 4e-5 | 4e-5 | 4e-5 |
| Batch Size | [2, 64] | 8 | 8 | 8 | 8 | 8 |
| $M_e$ | [20, 40] | 40 | 30 | 30 | 30 | 35 |
| $M_r$ | [20, 40] | 35 | 25 | 25 | 30 | 30 |
| $\alpha$ | [1e-5, 1e-2] | 1e-4 | 1e-4 | 1e-3 | 1e-4 | |
| $\beta$ | [1e-5, 1e-2] | 1e-3 | 1e-4 | 1e-4 | 1e-3 | |

Table 5: Configuration of Hyperparameters. $lr$ represents the initial learning rate. $M_e$, $M_r$ are the number of entity queries and the number of relation queries respectively. $\alpha$, $\beta$ are the weights of the bipartite consistency loss and entity-relation linking loss respectively.

| Method | Ent-F1 | Rel-F1 | sent/s |
|--------|--------|--------|--------|
| PFN (2021) | 66.8 | 38.4 | 55.1 |
| UniRE (2021) | 68.4 | 36.9 | 79.6 |
| PL-Marker (2022) | 69.9 | 41.6 | 18.5 |
| BiSPN (Ours) | 68.9 | 42.0 | 37.2 |
| w/o $\mathcal{L}_{ent}$ | 67.7 | 41.4 | - |
| w/o $\mathcal{L}_{link}$ | 68.0 | 41.3 | - |
| w/o $\mathcal{L}_{ent\leftrightarrow rel}$, $\mathcal{L}_{link}$ | 67.1 | 40.8 | - |

Table 6: Results on the SciERC dataset.