# OpenReview forum: "BiSPN: Generating Entity Set and Relation Set Coherently in One Pass"
_EMNLP/2023/Conference — EMNLP 2023 Findings_

### Official Review · Reviewer_6vNJ · 2023-07-28

**Soundness:** 4

**Excitement:**

3: Ambivalent: It has merits (e.g., it reports state-of-the-art results, the idea is nice), but there are key weaknesses (e.g., it describes incremental work), and it can significantly benefit from another round of revision. However, I won't object to accepting it if my co-reviewers champion it.

**Missing References:**



**Paper Topic And Main Contributions:**

The authors combined a set prediction network (SPN) for predicting entities and another SPN for predicting relation triplets. They integrated these two SPNs together and used consistency loss to enhance the predictive results.

**Reasons To Accept:**

- This article is well-written and very easy to follow, making it accessible for readers to understand the research and its contributions.

-The experimental results are promising, and the paper provides sufficient experiments and a detailed ablation study, which enhances the credibility of the proposed method and its performance evaluation.

**Reasons To Reject:**

- This article is relatively incremental. The author claims that using a set prediction network for the joint entity and relation extraction task is less explored. However, this simply involves combining the set prediction network for entity prediction and the set prediction network for relation prediction. The idea of adding a consistency loss is also natural and relatively trivial.

- Some commonly used datasets, such as ACE2004 and SciERC, were not included in the results presented by the authors. I want to know why the authors did not perform validation on these datasets as well. Additionally, many papers now provide results using the Rel+ metric. I would like to know why the authors did not include this metric in their evaluation

- Despite the promising experimental results, the proposed method does not demonstrate a significant advantage over the PL-marker; The authors should include the results of the PL-marker on the ADE dataset for comparison; Although it performs better in terms of speed compared to the two-pass model, it is significantly slower when compared to one-pass models such as UniRE.

**Reproducibility:**

4: Could mostly reproduce the results, but there may be some variation because of sample variance or minor variations in their interpretation of the protocol or method.

**Reviewer Confidence:**

4: Quite sure. I tried to check the important points carefully. It's unlikely, though conceivable, that I missed something that should affect my ratings.

---

> ### Author Rebuttal · Authors · 2023-08-26
>
> Thanks for your valuable feedback. We hope to address your concerns with the following responses:
>
> - Our work is an essential step to successfully apply set prediction networks on the joint extraction of entities and relations. Although the idea of adding a consistency loss is natural, it requires ingenious design and is not within easy reach.
>
> - We do not experiment on ACE2004 because it shares a similar corpus with ACE2005 and we believe ACE2005 is more representative than ACE2004. For the results on the SciERC dataset, please refer to our responses to Reviewer DE1H. In addition, the Rel-F1 metric used in our work (lines 372-377) is in fact the Rel+ metric that you mention, which is the most comprehensive metric for relation triple extraction.
>
> - As demonstrated by Table 3 in our paper, BiSPN achieves a better trade-off between accuracy and efficiency compared to the SOTA two-stage PL-marker and the SOTA one-stage method UniRE, which is a significant advantage of our method.
>
> - The results of PL-marker on the ADE dataset (shown below) will be included in a revised version of our paper. We can see that BiSPN outperforms PL-Marker by $0.5$ Rel-F1, while being competitive with PL-Marker in terms of Ent-F1.
>
> _A Comparison between PL-Marker and BiSPN on ADE_
>
> | Method       | Ent-F1 | Rel-F1 |
> | :----------- | :----: | :----: |
> | PL-Marker    |  92.2  |  83.2  |
> | BiSPN (Ours) |  92.1  |  83.7  |

---

### Official Review · Reviewer_w2i7 · 2023-08-05

**Typos Grammar Style And Presentation Improvements:** 1. Line 53
**Soundness:** 3

**Excitement:**

2: Mediocre: This paper makes marginal contributions (vs non-contemporaneous work), so I would rather not see it in the conference.

**Paper Topic And Main Contributions:**

This paper proposes a joint entity-relation extraction model, Bipartite Set Prediction Network (BiSPN). Specifically, BiSPN exploits a novel bipartite consistency loss and an entity-relation linking loss to maintain the coherence between the generated entity set and relation set. Experiments on biomedical/clinical datasets and a general-domain dataset show the effectiveness of BiSPN.

**Questions For The Authors:**

Please refer to "Reasons To Reject".

**Reasons To Accept:**

1. The motivation of this paper is intuitive.

2. This paper conducts extensive experiments and provides detailed analysis.

**Reasons To Reject:**

1. The generalization ability of the proposed model in the general domain appears to be limited.

2. It would be better to include an error analysis to further explore the limitations of the proposed model.

3. The writing of this paper needs to be further polished.

**Reproducibility:**

4: Could mostly reproduce the results, but there may be some variation because of sample variance or minor variations in their interpretation of the protocol or method.

**Reviewer Confidence:**

4: Quite sure. I tried to check the important points carefully. It's unlikely, though conceivable, that I missed something that should affect my ratings.

---

> ### Author Rebuttal · Authors · 2023-08-26
>
> Thanks for your valuable feedback. We hope to address your questions with the following responses:
>
> **Answer to Question 1:**
>
> Although our method does not surpass the SOTA two-stage method (PL-Marker) on the general domain dataset ACE05, it performs better than SOTA one-stage methods (PFN, UniRE) on the dataset. In this sence, the generalization ability of our method in the general domain should not be overlooked.
>
> **Answer to Question 2:**
>
> Because of the page limit of EMNLP, we mainly presented the main contributions of our paper, and did not conduct an error analysis. However, if the paper gets accepted for publication, we will include an error analysis to facilitate future research.
>
> **Answer to Question 3:**
>
> In spite of some typos, the overall writing of our paper is good, as suggested by other reviewers. And we do not think the writing of our paper prevents it from publication as we will fix these typos carefully.

---

### Official Review · Reviewer_DE1H · 2023-08-06

**Soundness:** 4

**Excitement:**

4: Strong: This paper deepens the understanding of some phenomenon or lowers the barriers to an existing research direction.

**Paper Topic And Main Contributions:**

This paper proposes a new method for joint entity and relation extraction. Their work is based on Set Prediction Networks (SPN) that uses a set of learnable queries that model the interaction between different relations/entities using attention mechanism. SPN has been applied to the two tasks of entity extraction and relation extraction separately, but this paper does that jointly. Their approach generates both entities and relations in one pass, but the challenge is to maintain the consistency between the two sets of outputs (e.g., entities in a triple should be in the list of extracted entities). They propose two auxiliary loss functions:

1. Bipartite consistency loss: For each sub/obj in a triple, it looks for a reference entity and encourages their representations to be similar to each other. Similarly, for each entity, it looks for an entity in the extracted triples and encourages their representation to be similar to each other.

2. Entity-Relation linking loss: This loss function encourages the representation of the entity queries and relation queries to be consistent. In particular, when a relation query and entity query are relevant (the gold entity associated with the entity query is present in the gold triple associated with the relation query), their representations are encouraged to be similar.

They perform experiments on ACE05 and three biomedical datasets. Their model works comparably or better than SOTA methods. Their methods achieves new SOTA on the three biomedical datasets, which are knowledge-intensive (multiple relations per sentence). However, their method gets worse results on ACE05. Through their experiments, they show that this is actually because ACE05 doesn't have many relations per sentence while their method is effective when more relations are involved in each sentence.

They perform important ablation studies showing that both proposed losses are useful. They also perform visualization and look into case studies to confirm their claim.

**Questions For The Authors:**

1. See reasons to reject 2.

2. Did you also consider other knowledge intensive datasets, e.g., do you know how the method works on SciERC?

3. Why should M_e be much larger than N_e (similarly for relations)?

**Reasons To Accept:**

1. This is a solid paper on joint entity/relation extraction with impressive results.

2. They have done a great job in explaining the details of the methods as well as the experiments.

3. The ablation studies, visualization, and case studies help confirm their claims.

4. They also show that their method is efficient compared to two-stage methods.

**Reasons To Reject:**

1. I'm worried about the generalizability of this method beyond relatively old/small LMs such as BERT. However, the smaller LMs lead to better efficiency which is one of the goals of the paper.

2. I couldn't find how exactly their method works during inference. Adding an explanation will help (how the many relation queries and entity queries are mapped to actual predictions).

3. The paper needs more proofreading throughout, but I'm sure this can be done in the updated version (see example typos below).

**Reproducibility:**

4: Could mostly reproduce the results, but there may be some variation because of sample variance or minor variations in their interpretation of the protocol or method.

**Reviewer Confidence:**

4: Quite sure. I tried to check the important points carefully. It's unlikely, though conceivable, that I missed something that should affect my ratings.

**Typos Grammar Style And Presentation Improvements:**

Figure 1: difficult to guaranteed => difficult to be guaranteed

Line 53: included the => included in the

101: all entity => all entities

---

> ### Author Rebuttal · Authors · 2023-08-26
>
> Thanks for your valuable feedback. Here are our responses to your questions:
>
> **Answer to Question 1:**
>
> As stated in lines 160-163, a null label ∅ is employed to indicate that no entity or no relation is found by an entity/relation query. When calculating the Prediction Loss, we assign the null label to entity/relation queries that are unassociated with gold entities/relations. During inference, the entity predictor filters out any entity whose predicted label is ∅ (lines 250-251) and the relation type predictor filters out relation triples similarly. In practice, the model additionally filters out any predicted entity or relation triple with boundary score below a predefined threshold.
>
> **Answer to Question 2:**
>
> By the time we submitted the paper, we had not yet experimented on the SciERC dataset due to time limit. We carried out experiments on SciERC afterward. The results are as follows:
>
> _Main results on the SciERC dataset_
>
> | Method                                                                    | Ent-F1 | Rel-F1 | sent/s |
> | :------------------------------------------------------------------------ | :----: | :----: | :----: |
> | PFN                                                                       |  66.8  |  38.4  |  55.1  |
> | UniRE                                                                     |  68.4  |  36.9  |  79.6  |
> | PL-Marker                                                                 |  69.9  |  41.6  |  18.5  |
> | BiSPN (Ours)                                                              |  68.9  |  42.0  |  37.2  |
> | w/o consistency loss                                             |  67.7  |  41.4  |   -    |
> | w/o linking loss                                                      |  68.0  |  41.3  |   -    |
> | w/o both                                                                  |  67.1  |  40.8  |   -    |
>
> In terms of entity recognition, BiSPN underperforms SOTA two-stage method PL-Marker by $1.0$ Ent-F1, but still outperforms SOTA one-stage methods (PFN, UniRE) substantially. In terms of relation triple extraction, BiSPN establishes new SOTA ($42.0$ Rel-F1) on the SciERC dataset. In terms of inference speed, BiSPN is about $2\times$ faster than PL-Marker, but slower than other one-stage methods. The results after ablating the consistency loss and the linking loss verify the effectiveness of them on the dataset.
>
> **Answer to Question 3:**
>
> In fact, for a specific corpus, $M_e$ ($M_r$) should be larger than the maximum number of entities (relation triples) occurring in a single sentence of the corpus, so that our method can be valid for all sentences of the corpus.

---

### Meta-Review · Area_Chair_iNFx · 2023-09-24

**Recommendation:** 3

**Metareview:**

This paper proposes a joint extraction method of entities and relation triples by combining set prediction networks (SPNs) and a new loss function that promotes global consistency. The effectiveness of the work is demonstrated on three biomedical datasets, where SOTA performance was achieved; however, their method didn't outperform baselines on ACE05. Through their experiments, the paper argues that this is because ACE05 doesn't have many relations per sentence while their method is effective when more relations are involved in each sentence. As pointed out by one of our reviewers, some commonly used datasets, such as ACE2004 and SciERC, were not included in the paper; many recent papers now use the Rel+ metric, which is also not presented in this paper.

The soundness of the work is well-perceived by all of the reviewers but reviewers split in terms of excitement. The general idea of joint extraction of entities and relations has been explored in the field for a long time, in training, in inference, or in both training and inference, so from that angle, the idea is not exciting enough. The novelty mainly resides only in the choice of SPN and the newly defined consistency loss function. The lack of analysis on other commonly used datasets and metrics is another limitation of the work.

---

### Decision · Program_Chairs · 2023-10-07

**Decision:**

Accept-Findings

**Comment:**

This paper proposes a joint extraction method of entities and relation triples by combining set prediction networks (SPNs) and a new loss function that promotes global consistency. The effectiveness of the work is demonstrated on three biomedical datasets, where SOTA performance was achieved; however, their method didn't outperform baselines on ACE05. Through their experiments, the paper argues that this is because ACE05 doesn't have many relations per sentence while their method is effective when more relations are involved in each sentence. As pointed out by one of our reviewers, some commonly used datasets, such as ACE2004 and SciERC, were not included in the paper; many recent papers now use the Rel+ metric, which is also not presented in this paper.

The soundness of the work is well-perceived by all of the reviewers but reviewers split in terms of excitement. The general idea of joint extraction of entities and relations has been explored in the field for a long time, in training, in inference, or in both training and inference, so from that angle, the idea is not exciting enough. The novelty mainly resides only in the choice of SPN and the newly defined consistency loss function. The lack of analysis on other commonly used datasets and metrics is another limitation of the work.